# Impact of COVID-19 on Life of Students: Case Study in Hong Kong

**DOI:** 10.3390/ijerph181910483

**Published:** 2021-10-06

**Authors:** Albert Lee, Vera M. W. Keung, Vincent T. C. Lau, Calvin K. M. Cheung, Amelia S. C. Lo

**Affiliations:** 1Centre for Health Education and Health Promotion, The Chinese University of Hong Kong, 4/Floor, Lek Yuen Health Centre, Shatin, Hong Kong, China; verakeung@cuhk.edu.hk (V.M.W.K.); vincentlau@cuhk.edu.hk (V.T.C.L.); calvincheung@cuhk.edu.hk (C.K.M.C.); amelialo@cuhk.edu.hk (A.S.C.L.); 2School of Public Health, Prince of Wales Hospital, Shatin, JC School of Public Health and Primary Care, The Chinese University of Hong Kong, 4/Floor, School of Public Health, Prince of Wales Hospital, Shatin, Hong Kong, China

**Keywords:** student health, health beliefs, health attitudes, healthy lifestyles, COVID-19, hygiene precautionary measures

## Abstract

COVID-19 has an impact on the day-to-day life of students, with school closure and detrimental effects on health and well-being that cannot be underestimated. A study collected data reflecting the health and well-being of secondary school students entering a programme entitled “Healthy Life Planning: Assist Students to Acquire and Practice Health Knowledge and Skills” (ASAP study) in September and October 2019 before the outbreak of COVID-19. Follow-up data were collected in June and July 2020, over half a year since the spread of COVID-19, which facilitated analyses of its impact on the health behaviours and well-being of young people. Comparative analyses between baseline and the follow-up period were conducted on weight status, sleep pattern and quality, pattern of sedentary lifestyle, pattern of physical activity, attitudes and perceived barriers for exercise, and hand hygiene. Attitudes toward precautionary measures and influenza vaccination, self-reported changes in hygiene practices, exercise habits and eating habits were analysed. Although hygiene habits and risk perceptions among young people have improved in many aspects, the level of physical activity has declined as well as the beliefs and attitudes on increasing time on electronic media and change in sleep hygiene. Attitudes and beliefs towards influenza vaccination have declined, which would reflect the slow increase in the uptake rate of COVID-19 vaccination. Health education should equip students with the knowledge and skills to cultivate beliefs and attitudes to face health challenges.

## 1. Background and Introduction

Since the COVID-19 pandemic was declared, lockdown measures have been implemented in many parts of the world. Implementation of physical measures to interrupt or reduce the spread of respiratory viruses based on sustained physical distancing, restriction of social gathering, and “shut-down” measures has a strong potential to reduce the magnitude of the peak of the COVID-19 pandemic [1].

However, impacts on other aspects of health must not be underestimated. A study during the semi-lockdown period has shown males with BMI 24 or above had lost weight, but all other subjects had gained weight as a result of a significant decline in the amount of moderate or vigorous exercise [2]. Obesity has been shown to increase the risk of mortality of COVID-19 after adjusting for confounding factors such as age in different parts of the world [3].

A study by Fong et al. in 2020 found that 65.3% of participants experienced increased stress due to staying at home and 29.7% experienced moderate to severe levels of depressive symptoms; increases in the use of electronic devices and decreases in outside activities were positively associated with a higher level of depression severity [4]. Studies have also found increasing prevalence of obesity [5] and myopia [6] among school children due to longer screen times, lack of physical activity, and living in small, crowded living and learning spaces at home. Increasing physical activity and maintaining a healthy diet, leading to positive changes to their physical health, have also been shown to be associated with better mental health [7,8]. Non-communicable diseases such as cardiovascular diseases, chronic lung diseases, cancer and diabetes are still constituting the main health burdens of society [9]. The main drivers for an unhealthy diet and lack of physical activity would be a lack of places and opportunities to be physically active and industries’ opposition to public health interventions [10]. Behavioural, environmental and occupational, and metabolic risks can explain half of the global mortality and more than one-third of global disability-adjusted life year (DALY) [11]. A substantial burden of global cardiovascular disease morbidity and mortality is attributable to a sedentary lifestyle, and the attributable burden of high BMI has increased in the past 23 years; physical inactivity and unhealthy eating are the key underlying causes [11].

COVID-19 also has an impact on the day-to-day life of students with school closures [12]. Results of one study have shown a dramatic decline in assessment during COVID-19 in schools, suggesting lower performance when students start school in 2020 [13]. Schools may need to leverage decision-making frameworks, such as the Multi-Tiered Systems of Support/Response-to-Intervention (MTSS/RTI) framework [14] to identify needs and target instruction where it matters most when school begins in late 2020. During the first half of the academic year in 2020 in Hong Kong, schools were closed during the spring term with online learning, with half-day sessions in the summer term before closure again due to a third wave in July 2020 in Hong Kong. Schools reopened after the summer break in September 2020, with half-day sessions, and closed again in early December 2020 due to the fourth wave. Schools reopened in February 2021 with half-day sessions. The government has imposed restrictions on social gathering including numbers of people grouped together and the operation of restaurants and recreation facilities. Many recreation facilities including public utilities were closed or operated under strict control of people flow periodically in 2020. There is a need to study the impact of COVID-19 on student life with disruption of usual school life and social interaction during that period.

The Centre for Health Education and Health Promotion of the Chinese University of Hong Kong (CHEHP) has pioneered the Healthy School/Health Promoting School (HPS) movement in Hong Kong and neighbouring countries over the last two decades [15,16]. It has developed many initiatives, making use of the HPS framework to improve the health literacy of students [15]. Recently, it launched the ASAP study (Healthy Life Planning: Assist Students to Acquire and Practice Health Knowledge and Skills) to enrich the knowledge and skills of students on a variety of health-related matters. The ASAP Project provided health educational materials covering nine teaching units designed for junior secondary schools. Topics covered sleep hygiene, infectious disease control, travellers’ health, physical activity, body image, stress management, etc. From these, teachers chose one or more units for school-based curriculum enrichment. They were also required to develop experiential learning activities for students based on the topics they have taught. Students might conduct project learning on them as well.

The impact of COVID-19 on the lives of students who received the ASAP program is being studied. The aim of this study is to investigate the impact of COVID-19 on student health and well-being by collecting data reflecting the health and well-being of students at the entry of ASAP (before COVID-19 outbreak), then at a yearly interval (after the outbreak), to analyse any changes. 

## 2. Materials and Methods

### 2.1. Study Design

For this case study, comparative analyses between baseline and follow-up periods were conducted to identify potential changes in students’ weight status, sleep pattern and quality, pattern of sedentary lifestyle, pattern of physical activity, attitudes and perceived barriers for exercise, and hand hygiene. The attitudes toward precautionary measures to COVID-19 and influenza vaccination, self-reported changes in daily living habits, exercise habits, eating habits and hygiene practices were analysed.

The study has been approved by the Survey and Behaviour Research Ethics Committee (SBRE-19-104). The surveys were anonymous. The participating schools have obtained consent from parents and students and students’ participation was entirely voluntarily with no adverse repercussions. 

### 2.2. Study Population

The study targeted students studying in grades between Secondary 1 (S1) and Secondary 3 (S3), aged about 11–15 years.

### 2.3. Sample Population

Eleven secondary schools in Hong Kong that participated in the ASAP study were invited to the pre-and-post questionnaire survey. School teachers were allowed to use the teaching materials provided by the programme to enrich their health-related curricula such as Physical Education, Technology and Living, Biology, and the school-based health curriculum. The teaching materials covered various health contents such as physical activities, sleep hygiene, stress management, body image, infectious disease control, dental health, the prevention of prolonged use of electronic devices, etc. At least one grade between S1 and S3 of the participating schools was beneficial to the study and eligible for the survey. A total of 1355 students studying in the selected grades were invited, and 1102 completed two administrations of the questionnaire, giving a response rate of 81.3%. The survey was anonymous and used the class number of each responding student to match the questionnaires completed in two administrations in September and October 2019 (baseline) and June and July 2020 (follow-up), respectively.

### 2.4. Measuring Tools

The Hong Kong Student Health Survey Questionnaire (HKSHQ) was used to collect data reflecting lifestyles, including hygiene practice and general health status. HKSHQ adopts a system of surveillance of student health status, taking reference from the US Centres for Disease Control and Prevention (CDC) Youth Risk Behavioural Surveillance (YRBS) [17,18] and Wessex Healthy School Award [19], which has been used by CHEHP [20,21,22] with continuous refinement as a tool for assessing student health status and health-related outcomes [16].

The parameters on demography include date of birth, gender, and self-rated health status (3 questions). The survey also uses the Family Affluence Scale (FAS), which was utilised to reflect the economic status of the respondents’ family from the following criteria (4 questions): the number of vehicles owned by the respondent’s family; whether the respondent has a separate bedroom; the number of family trips; and the number of computers owned by the family [23,24]. The Pittsburgh Sleep Quality Index was utilised to measure sleep quality [25]. 

Self-reported body weight of students was classified into wasting, desirable, and obese according to the weight-for-height charts in a local guide to childhood growth and nutrition assessment by Leung [26]. The charts are gender-specific, in which obesity is defined as body weight values above 120% of the median weight-for-height, while wasting is defined as body weight values below 80% of the median weight-for-height. When the body height value of a subject exceeds the data available in the charts, Body Mass Index (BMI) cut-offs for Asian adult populations are used to interpret the subject’s body weight, where a value between 18.5 kg/m^2^ and 22.9 kg/m^2^ is considered normal [27]. 

The Theory of Planned Behaviour by Ajzen [28] was applied in this survey to assess the attitudes and perceived behavioural control on physical activity (Appendix A). Similarly, the study also assessed the attitudes and perceived behavioural control on the uptake of influenza vaccination. COVID-19 vaccination was not available at the time of data collection, so their attitudes towards influenza vaccination would help us understand their perspectives on vaccination. Since follow-up data were collected during the COVID-19 pandemic, questions reflecting the respondents’ risk perception (such as the wearing of face masks, hand hygiene, social distancing, actions taken with suspected symptoms) were added to the questionnaire.

The Rosenberg Self-esteem Scale (RSE) by Morris Rosenberg was adopted in this survey to evaluate self-esteem in teenagers at the baseline of the study [29,30]. Leung and Wong [31] studied the validity and reliability of the Chinese translation of the RSE and gave recommendations on the Chinese wordings in some of the items. The current study adopted the Chinese translations recommended by Leung and Wong for item 3 (“I feel like a person who has a number of good qualities”), 7 (“I feel that I am a person of worth, at least on an equal plane with others”) and 8 (“I wish that I could have more respect for myself”) to optimise the reliability. The RSE and the Theory of Planned Behaviours [28] give a complete description of the non-cognitive development of the participants and a clear indication of the effects of the interventions in developing the habit of doing exercise and receiving a flu vaccine to prevent them from being infected.

The Mental Toughness Scale for Adolescents (MTS-A) by McGeown, St. Clair-Thompson and Putwain [32] was adopted in this survey to examine the mental toughness of teenagers before and after the interventions. The scale is an 18-item Likert scale with items answered on a four-point scale from “strongly disagree” to “strongly agree”. The concept of mental toughness in adolescents includes six domains: challenge, interpersonal confidence, confidence in abilities, emotion control, control of life, and commitment. Three statements describe each of the above domains in the teenager context, and respondents have to indicate how strongly they agree or disagree with each sentence. The author of MTS-A has granted the research team permission to use the scale supplemented with the Chinese translation. 

### 2.5. Data Collection

The study collected data reflecting the health and well-being of students at the beginning and then at a yearly interval to monitor any changes. The baseline data were collected in September and October 2019 at the beginning of the academic year before the outbreak of COVID-19, and follow up data were collected in June and July 2020, half a year after its outbreak.

### 2.6. Data Analysis

The McNemar test was used to determine if there were differences among dichotomous dependent variables (such as whether the subjects had played ball games over the last seven days) between pre and post groups. Paired t-test was used for similar purposes but for comparing the means of continuous dependent variables (such as the subjects’ attitude score toward physical activities). A difference was considered statistically significant if the *p*-value was <0.05. Data were analysed by SPSS Statistics, version 25.0.

## 3. Findings

Table 1 describes the background demographic characteristics of the subjects, including socioeconomic status. The subjects had an average age of 13.28 years at baseline and 13.99 years at follow-up (standard deviation: 1.07 year). Sixty percent (60.2%) of them were female because two participating schools were girls’ schools, while the other nine were co-education. The subjects came from schools in urban settings, semi-urban settings and satellited towns. 

About 50% of students came from the middle affluence group and about one-quarter from either high or low affluence groups. Most of the schools in this study are located in districts with monthly median domestic household incomes below the overall median level in Hong Kong. The sample is not skewed towards higher socioeconomic groups. 

Results of the current study show that the proportion of students classified as obese decreased from 23.0% to 20.5% and 13.3% to 12.0% among male and female students, respectively. The changes were not statistically significant.

The percentage of students engaged in 60 min of moderate to vigorous exercise decreased with statistical significance from 40.8% to 30.1%, particularly those rigorous activities taking place in groups or in public, or vigorous activities such as running and jogging, ball games, swimming, playground activities, skating, and martial arts (Table 2). The item “stretching” was added to the post-test questionnaire. Over one-fourth of students (26.6%) reported that they had done some stretching during the seven days before the post-survey, but no baseline data were available for direct comparison. 

Higher proportion of students spent more than two hours on an average school day watchng video programmes as well as internet surfing (not for academic purpose) on both ordinary school days and during holiday with statistical significance (Table 3). The percentage of students who perceived no influence on the prolonged use of electronic media increased, and those who perceived eye fatigue and shoulder discomfort reduced (Table 3). However, an increased impact on their concentration and study was reported with statistical significance (Table 3). The proportion of students going to bed after 11:00 pm increased from 43.5% to 66.1%, and that of students getting up after 8:00 am increased from 10.0% to 32.9% with statistical significance, though sleep quality was not affected significantly (Table 3). Self-reported handwashing behaviours improved, with a higher proportion of students washing hands thoroughly and a smaller proportion not taking handwashing seriously with statistical significance (Table 4).

Table 5 shows the changes in attitudes and beliefs towards physical activities from baseline to follow-up. The decline is observed in the goal of action, attitudes, subjective norm, perceived behavioural control, behavioural beliefs and norm beliefs with statistical significance. The behavioural intention and control beliefs also declined, although statistical significance was not detected. 

Regarding the changes in attitudes and beliefs towards influenza vaccination from baseline to follow-up, Table 6 shows a decline in all domains with statistical significance, particularly behavioural intention and subjective norm and perceived behavioural control. Students are a target group for influenza vaccination in Hong Kong. Table 7 shows that a high proportion of students would continue wearing face masks and handwashing, but there was a lower proportion for other hygiene measures. This is reflected by just over half of students (54.9%) reporting a significant change in hygiene habits. More than half of students (52.8%) reported a decrease in physical activities such as running and walking, and 41.2% reported fewer ball games, and only a low proportion of students reported having participated in other physical activities such as outdoor activities (Table 7). Although students tend to eat healthier at home, this proportion (55.0%) is not very high, and less than one-fifth of students (17.5%) had a significant change in eating habits (Table 7).

Table 8 shows students’ intention to maintain precautionary measures over the next three months post-test. The majority of students would continue to wear a face mask and be meticulous about handwashing, in line with findings of current practices, shown in Table 6. About half of the students would like to see a relaxation on physical distancing and restriction of gathering to allow more interaction. Students have a higher risk perception of respiratory symptoms; they would not go to school or activities and would only continue if no fever and reporting symptoms (Table 8).

## 4. Discussion

The decline in the level of physical activity and the prolonged use of electronic media, with increasing effects on students’ learning, concentration, and sleep pattern (going to bed late and getting up late), are worrying (Table 2 and Table 3). Socioecological models state that a person’s health status is not only influenced by individual behaviours, but also by factors situated in a person’s environment [33,34]. The concept of “environment” captures multiple dimensions, and a Built Environment (BE) can be defined broadly as “the human-made space in which people live, work and recreate on a day-to-day basis” [35]. During the COVID-19 pandemic, the BE has been altered due to various preventive and lockdown measures. It not only encompasses green spaces and parks, but also includes the internal environment and social capital (defined as social networks and interactions that inspire trust and reciprocity among citizens) [36]. The social environment, part of the BE, refers to factors such as social support and social networks, social deprivation, and social cohesion and systems [37]. BE shapes individual health behaviour through diverse mechanisms and can be adverse or beneficial for health [38]. Neighbourhoods that are more walkable, either leisure-oriented or destination-driven, are associated with increased physical activity, increased social capital, lower overweight rates, lower reports of depression, and less reported alcohol use [39]. Better street connectivity or walkability tended to be positively related to increased physical activity and walking [40]. 

One study has found that adolescents undertook more physical activity during lockdown if they had stronger prior physical activity habits, but some were unsure of what to do when they did not have instruction from a coach. Some adolescents reported that physical activity became a method of entertainment during lockdown, and this mindset change increased the level of physical activity [41]. Living space is very limited in Hong Kong, making physical activity at home not feasible for many young people. Online coach-led physical activity sessions have helped encourage and support adolescents to follow online exercise routines [41]. The implementation of lockdown measures and school closures has a significant impact on the BE, not only in terms of walkability and connectivity but also in terms of social connectivity and support. Apart from the effect on physical activities, we must not underestimate its negative effect on other aspects of health, such as psycho-social well-being, as a result of the impact of COVID on the BE diminishing social capital. This might be reflected by less positive beliefs and attitudes towards physical activities (Table 5). Around half of the students reported a decreased frequency of walking or running and ball games without much increase in other types of indoor physical activities (Table 7). 

Although staying at home should enable students to eat healthier, this proportion is not high and less than 20% of students had a significant change in eating habits (Table 7). Previous studies have revealed a low level of physical activities and healthy eating among secondary students [42,43]. COVID-19 might have worsened these conditions.

Some previous studies stated that lockdown and school closures might exacerbate childhood obesity [44] and cause unhealthy changes to the diet of students [45,46]. Past studies also support the claim that when students are not in school, they tend to have less healthy diets [47]. The findings of our survey showed similar results, with 29.2% students consuming unhealthy takeaway food, and one-fifth of students having increased consumption of soft drinks (20.2%), desserts (19.8%) and crispy food (19.7%). However, over half of the students (55.0%) indicated that they had healthier meals at home, and 38.6% of them consumed more fresh fruits, implying that the COVID-19 pandemic might have brought not only negative impacts but also some positive changes to the diet of students. Such positive changes may partly be explained by the fact that before the pandemic, most secondary students in Hong Kong consumed their lunch at nearby restaurants or fast food shops when they had whole-day classes on average school days [14]. School suspension as well as the fear of infection drove students to stay home for food, while lockdown and work-from-home arrangements also allowed more parents to prepare meals for their children. Further studies are required to investigate whether such changes will lead to any changes in childhood obesity in Hong Kong.

The percentage of students who perceived no influence on the prolonged use of electronic media increased, but those who perceived eye fatigue and shoulder discomfort reduced (Table 3). This may be due to adaptation. However, prolonged use had an impact on their studies and concentration as well as sleep pattern (Table 3).

It is encouraging to observe the improvement in hand hygiene reflected by more serious handwashing (Table 4). However, it is disappointing and alarming to find the decline in beliefs and attitudes, including motivation and perceived control, towards influenza vaccination with statistical significance (most showing p-value lower than 0.001) (Table 6). This could be due to school suspension during the pandemic, and so, they perceived having a lower risk of being infected. However, the scores at baseline were already low, which makes it difficult to identify a further significant decline. This might reflect the weak perception of the beneficial effect of influenza vaccination. It might also account for the slow increase in the uptake of COVID-19 vaccination in Hong Kong [48], which is also observed in other parts of the world [49]. Previous studies on predictive factors of influenza vaccination suggested that factors related to health belief models such as perceived adverse effects and efficacy and advice given by health care professionals are determinant factors for the uptake of vaccination [50,51].

The uptake rate of COVID-19 vaccines in Hong Kong is still unsatisfactory, despite the availability and accessibility of the vaccine. There is room for improvement to enhance the health beliefs and attitudes towards vaccines for preventing the disease. A study on the acceptance of the COVID-19 vaccine found that people who perceived the seriousness of the infection, vaccine conferring benefits, and received calls to action were significantly more likely to accept the vaccine [52]. Conversely, perception of barriers to accessibility and potential harm of the vaccine were found negatively to be associated with their acceptance. Recommendation by the government stood out as the most important cue. Public health intervention programmes focusing on increasing the perception of the benefits of vaccination and perceived susceptibility to infection while reducing the identified barriers should be warranted [53]. The study also revealed that the public values efficacy and safety more than the cost of vaccines. Another study in the US found that a greater likelihood of COVID-19 vaccine acceptance was associated with more knowledge about vaccines, less acceptance of vaccine conspiracies, elevated COVID-19 threat appraisals, and being up to date with influenza immunisation [49]. The other demographic predictors of a likelihood of being vaccinated against COVID-19 were higher income group (income of USD 120,000 or higher) and being a Democrat (in comparison to the reference category Republican), and respondents relying on social media for information about COVID-19 anticipated a lower likelihood of COVID-19 vaccine acceptance. More public health interventions targeting those factors facilitating and hindering uptake should be put in place.

The closure of schools during COVID-19 could result in the loss of opportunity to foster positive beliefs and attitudes in students towards influenza vaccination. It could also have an impact on the low uptake rate of COVID-19 vaccination. From the findings of this study, there is room to enhance the perception of the benefits of vaccination against infectious disease in students, particularly before pandemics and the potential consequences if not vaccinated. Health education should cultivate a positive and supportive culture to support family members and friends to receive the vaccination. Health literacy includes access and analysing health information and problem solving such as breaking the barriers to access these services. This would help to improve the acceptance and uptake rate. A recent study in Hong Kong has found a higher level of vaccine acceptance among the youngest adult group (age 18 to 24), which would be due to better exposure to vaccine education and receiving the free vaccine at birth [52]. Findings from this study have shown that students perceived the importance of wearing face masks in public places, were meticulous about handwashing and highly vigilant with regard to respiratory symptoms (Table 8). Risk perceptions are a critical determinant of health behaviour, and the profile of risk perceptions and accuracy of perception would affect the association between risk perceptions and health behaviours [54]. Although a high level compliance of facemask wearing was observed and more people maintained social distancing and used alcohol hand rub during the pandemic, decreasing willingness to accept the COVID-19 vaccines was also observed. This might be associated with increasing concerns about vaccine safety and growing compliance of personal protection behaviours [55]. Therefore, the concept of “ASAP” should be adopted for school curriculum development to assist students in acquiring and practicing health knowledge and skills, including health risk perception and preventive measures for infectious diseases from a broader perspective that includes vaccination. 

A substantial proportion of students expressed their wishes to relax social distancing and restriction of gathering (Table 8). Although measures such as closing and restricting most places where people gather in smaller or larger numbers for extended periods (businesses, bars, schools and so on) are most effective, they can cause substantial collateral damage to society, the economy, trade and human rights [56]. This study has shown the collateral damage to students’ health and well-being and their health beliefs and attitudes. The COVID-19 pandemic has also been found to lead to an increase in myopia among young children in Hong Kong; the prevalence of myopia among school-age children during the pandemic has increased significantly compared to a study conducted before the outbreak [57]. Prolonged exposure to screens and less time spent outdoors were linked to faster progress in myopia, according to researchers. One study found several highly effective measures that are less intrusive, including land border restrictions, governmental support to vulnerable populations and risk-communication strategies [58]. Therefore, governments and other stakeholders should consider adopting non-pharmaceutical interventions tailored to the local context when infection numbers surge (or surge a second time) before choosing those intrusive options. Less drastic measures may also foster better compliance from the population [52].

There are limitations to this study. The subjects are participants of the ASAP study, not a random sample of secondary students. The demography of the students is not markedly different from the demography of students in Hong Kong. They do not skew towards particular demographic characteristics except for the subjects’ gender as two schools are girls’ schools while the others are co-education.

There is a potential bias that they are more health-conscious and have better knowledge and more positive attitudes towards health. Most of the schools are located in districts with median monthly household income below the median in Hong Kong. The sample is not skewed towards higher socioeconomic groups. The students should be more resilient towards the impact of COVID-19 on healthy living. The findings of the study that reflect the beliefs, attitudes, perceived control, and behaviours of students under the pandemic have significant implications. There is an assumed hypothesis that students with better health literacy will maintain positive health beliefs and positive attitudes and behaviours towards healthy living. The findings will help to test this assumption and shed light on which aspects of their beliefs, attitudes and behaviours can be sustained under adverse conditions (such as COVID-19) and how young people should be supported further, notwithstanding that they might have enriched knowledge and skills in health.

Another limitation is the lack of a control group. It is technically difficult to engage more students and schools to participate in the survey under the COVID-19 situation. Moreover, there will not be a perfect control group as schools and students cannot be controlled to receive information and skills enhancement to fight against COVID-19. However, the study has included studies on belief, perceived barriers of control, and attitudes. The findings would partially explain why students behave in a particular way during the COVID-19 period. The global impact of the COVID-19 pandemic has not been experienced for nearly a century. Data reflecting the impact on students’ life would provide useful insights for combating similar challenges in the near future.

## 5. Conclusions

The current study reveals the changes in physical activities, hygiene and dietary behaviours in Hong Kong adolescents between September 2019 and July 2020, when the novel coronavirus disease (COVID-19) started to hit many parts of the world, resulting in the pandemic. These changes include less moderate and rigorous physical activities, and the attitudes and beliefs of students towards physical activities have become less positive and less persistent. Although hygiene habits and risk perceptions among young people have improved in many aspects, attitudes and beliefs towards influenza vaccination have declined, which would reflect the slow increase in the uptake rate of COVID-19 vaccination. This study has shown the changes in students’ health behaviours, beliefs and attitudes. Health education targeting young people and the public should equip them with the knowledge and skills to cultivate beliefs and attitudes and this would have impact on risk perceptions and behaviours to face health challenges.

## Figures and Tables

**Table 1 ijerph-18-10483-t001:** Demographic characteristics of the subjects (N = 1102).

	Percentage (Number of students participated in the study) [ ] Monthly Median Domestic Household Income by Census 2016 USD 1 = HKD 7.8
Gender:	
Male	39.8% (439)
Female	60.2% (663)
Grade:	
Secondary 1	50.8% (560)
Secondary 2	17.2% (189)
Secondary 3	32.0% (353)
Overall Hong Kong Monthy Median Domestic Income [HKD 25,000]	
Socioeconomic status based on the Family Affluence Scale as an indicator:	
Low affluence group	25.2% (272)
Middle affluence group	51.5% (556)
High affluence group	23.3% (252)
Location of participating schools in Hong Kong:	
Tuen Mun (3 schools) ^a^ [HKD 22,000]	22.1% (243)
Sham Shui Po (2 schools) ^b^ [HKD 20,000]	22.5% (248)
Kwun Tong (1 school) ^b^ [HKD 20,160]	12.6% (139)
Yau Tsim Mong (1 school) ^b^ [HKD 23,500]	10.5% (116)
Kwai Tsing (1 school) ^c^ [HKD 21,600]	9.8% (108)
Shatin (1 school) ^c^ [HKD 27,180]	8.0% (88)
Kowloon City (1 school) ^b^ [HKD 25,500]	7.9% (87)
Sai Kung (1 school) ^a^ [HKD 32,470)	6.6% (73)

^a^ Semi-urban setting ^b^ Urban setting. ^c^ Satellite towns (evolved from rural areas to urban setting).

**Table 2 ijerph-18-10483-t002:** Level of physical activity.

	Percentage of Students at Baseline (Number)	Percentage of Students at Follow up (Number)	Number of Valid Cases	*p*-Value
**Time Spent on Physical Activities**				
60 min moderate to vigorous exercise ≥3 days over last 7 days (↓)	40.8% (442)	30.1% (325)	1081	<0.001
**Types of Activities Engaged Over Last 7 Days**				
Running and jogging (↓)	52.0% (558)	36.0% (387)	1074	<0.001
Ball games (e.g., basketball, soccer, badminton, volley ball) (↓)	40.0% (430)	20.7% (222)	1074	<0.001
Swimming (↓)	12.9% (139)	5.5% (59)	1074	<0.001
Group game activities (↓)	10.4% (112)	3.1% (33)	1074	<0.001
Playground activities (↓)	7.5% (81)	2.2% (24)	1074	<0.001
Martial Arts (↓)	5.9% (63)	1.6% (17)	1074	<0.001
Skating (↓)	4.7% (51)	2.0% (22)	1074	<0.001
Physical training (e.g., going to the gym) (↓)	8.5% (91)	6.3% (68)	1074	0.045
Dancing/gymnasium	11.5% (124)	11.7% (126)	1074	0.925
Electronic physical games	9.1% (98)	8.8% (95)	1074	0.867
Rope skipping	7.6% (82)	6.7% (72)	1074	0.382
Hiking/outdoor walk	5.6% (60)	7.4% (79)	1074	0.096
Cycling	7.5% (80)	6.3% (68)	1074	0.251

*Footnote*. The item “stretching” was added to the post-test questionnaire. Over one-fourth of students (26.6%) reported that they had done some stretching during the seven days before the post-survey, but no baseline data were available. McNemar Test was performed. Arrows indicate the direction of significant changes. NS: non-significant.

**Table 3 ijerph-18-10483-t003:** Time spent on electronic media (non-academic purpose) and sleep time.

	Percentage of Students at Baseline (Number)	Percentage of Students at Follow up (Number)	Number of Valid Cases	*p*-Value
**Time Spent on Electronic Media over 2 h Every Day**				
Television, YouTube and TV online on an average school day (↑)	50.2% (540)	56.8% (611)	1076	<0.001
Television, YouTube and TV online during holiday	72.3% (778)	74.9% (806)	1076	0.123
Electronic and Computer games on an average school day	39.0% (421)	41.9% (452)	1080	0.100
Electronic and Computer games during holiday	60.1% (643)	62.6% (670)	1070	0.175
Internet surfing on an average school day (↑)	27.2% (295)	38.1% (414)	1086	<0.001
Internet surfing during holiday (↑)	39.1% (422)	48.4% (522)	1079	<0.001
**Perceived Impact on Prolonged Time on Electronic Media**				
No perceived impact at all (↑)	37.8% (409)	47.4% (512)	1081	<0.001
Eye fatigue (↓)	41.0% (443)	33.6% (363)	1081	<0.001
Effect on study (↑)	16.5% (178)	21.5% (232)	1081	0.001
Decline of concentration (↑)	14.8% (160)	19.3% (209)	1081	0.001
Inadequate sleep leading to fatigue (↓)	19.8% (214)	16.8% (182)	1081	0.036
Shoulder discomfort (↓)	15.6% (169)	12.1% (131)	1081	0.007
Tension with family (↓)	15.8% (171)	12.7% (137)	1081	0.016
Emotion fluctuation	8.9% (96)	9.3% (101)	1081	0.748
Back discomfort	9.3% (100)	9.7% (105)	1081	0.733
Hand discomfort	8.1% (88)	7.4% (80)	1081	0.539
**Sleeping Time**				
Sleep after 11:00 pm (↑)	43.5% (471)	66.1% (716)	1083	<0.001
Waking up after 8:00 am (↑)	10.0% (109)	32.9% (360)	1094	<0.001
Average sleep hour ± standard deviation (↑)	7.75 ± 1.47	7.93 ± 1.87	1079	0.004
**Sleep Quality** (mean ± standard deviation of PSQI)				
Average score ± standard deviation	4.81 ± 2.61	4.87 ± 2.59	1018	0.470

*Footnote*. McNemar Test was performed except for comparing the average sleep hours and the scores of Pittsburgh Sleep Quality Index (PSQI). A PSQI score above 5 indicates poor sleep quality in the respondent. Paired t-test was performed to compare means. Arrows indicate the direction of significant changes. NS: non-significant.

**Table 4 ijerph-18-10483-t004:** Self-reported handwashing behaviours (number of valid cases = 971).

	Percentage of Students at Baseline (Number)	Percentage of Students at Follow up (Number)	*p*-Value
Washing hands meticulously with adequate soap over different positions, including the back of the hand, wrist, gaps between fingers (↑)	14.7% (143)	22.2% (216)	<0.001
Washing hands with soap over different positions, including the back of the hand, wrist, gaps between fingers but not meticulously (↑)	37.9% (368)	45.2% (439)	<0.001
Washing hands quickly, not always with soap (↓)	38.1% (370)	26.1% (253)	<0.001

*Footnote*. McNemar Test performed. Arrows indicate the direction of significant changes.

**Table 5 ijerph-18-10483-t005:** Attitudes and beliefs toward physical activities.

Domain (number of item)	Content	Range of scores	Average score at baseline (±SD)	Average score at follow up (±SD)	Number of valid cases	*p*-value
Goal of action (1 item)	Number of days in 7 days that I can perform moderate to vigorous physical activity for 60 or more minutes	0 to 7	2.38 (±2.01)	1.88 (±2.03)	1081	<0.001
Behavioural intention (1 item)	Intend to put more efforts in doing physical activity in the next 2 weeks	−3 to 3	−0.46 (±1.80)	−0.54 (±1.76)	1049	0.159
Attitudes (4 items)	Being positive towards doing physical activity	−3 to 3	0.85 (± 1.41)	0.63 (±1.36)	1038	<0.001
Subjective norm (2 items)	Friends perform exercise regularly	−3 to 3	0.10 (±1.50)	−0.03 (1.43)	1066	0.005
Perceived behavioural control (2 items)	Doing 60 min exercise every day can be achievable over the next 2 weeks	−3 to 3	−0.06 (±1.55)	−0.24 (±1.47)	1066	<0.001
Behavioural beliefs (4 items)	Exercise makes me feel more healthy	−36 to 36	12.26 (± 12.98)	11.30 (±12.60)	1047	0.022
Norm beliefs (2 items)	Health experts think that I should do more exercise	−18 to 18	3.82 (± 5.87)	3.25 (±5.69)	1032	0.011
Control beliefs (2 items)	I have spare time to do physical activity	−42 to 42	10.39 (±14.80)	9.59 (±13.80)	1043	0.081

*Footnote*. Paired t-test was performed to compare means. NS: non-significant.

**Table 6 ijerph-18-10483-t006:** Attitudes and beliefs toward influenza vaccination.

Domain (number of item)	Content	Range of scores	Average score at baseline (±SD)	Average score at follow up (±SD)	Number of valid cases	*p*-value
Behavioural intention (1item)	I will get vaccinated before the next flu epidemic	−3 to 3	0.65 (break)(± 1.91)	0.45 (±1.82)	1055	0.002
Attitudes (4 items)	Vaccination will be beneficial to me	−3 to 3	0.82 (±1.43)	0.71 (±1.37)	1035	0.023
Subjective norm (2 items)	People important to me want me to get vaccinated	−3 to 3	0.62 (±1.59)	0.29 (±1.62)	1046	<0.001
Perceived behavioural control (2 items)	Getting vaccinated before the flu epidemics is easy to me	−3 to 3	0.54 (±1.34)	0.36 (±1.20)	1037	<0.001
Behavioural beliefs (2 items)	Vaccination will lower my risk of getting a flu	−18 to 18	4.38 (±5.81)	3.66 (±5.89)	1027	0.001
Norm beliefs (2 items)	The family wants me to get vaccinated	−18 to 18	4.81 (±7.06)	3.70 (±6.18)	946	<0.001
Control beliefs (1 item)	School or clinics provide the information and services	−21 to 21	5.85 (±8.21)	4.70 (±7.61)	977	<0.001

*Footnote*. Paired t-test was performed to compare means.

**Table 7 ijerph-18-10483-t007:** Change in health and hygiene behaviours during COVID-19.

Behaviours	Percentage of Students (Number)
**Hygiene behaviours (number of valid cases who reported modest to significant changes in hygiene habits during the pandemic = 894)**	
Increased use of face mask in public place	92.4% (826)
Increasing frequency of handwashing	80.8% (722)
Covering toilet when flushing	59.6% (533)
More meticulous in following the steps of handwashing	55.9% (500)
Frequent change of clothing	49.6% (443)
Reduced frequency of rubbing eyes, nose and mouth	48.0% (429)
More meticulous in cleaning body during bathing	43.7% (391)
More frequent in cleaning the house	39.9% (357)
**Self-reporting change in hygiene habits (Total number = 1087)**	
Reporting significant change in hygiene habits	54.9% (597)
Reporting modest change in hygiene habits	27.3% (297)
**Physical activity (number of valid cases who reported modest to significant changes in exercise habits during the pandemic = 651)**	
Decreased frequency of running and walking	52.8% (344)
Less ball games	41.2% (268)
More stretching exercise at home	37.9% (247)
Decreased water sport	17.8% (116)
Increased going to the countryside or hiking	16.0% (104)
Decreased going to the countryside or hiking	10.8% (70)
Decreased dancing activities or martial arts activities	9.4% (61)
**Self-reporting change in exercise habits (Total number = 1089)**	
Reporting significant changes in exercise habits	24.2% (263)
Reporting modest change in exercise habits	35.6% (388)
**Eating behaviours (number of valid cases who reported modest to significant changes in eating habits during the pandemic = 655)**	
Increased frequency of dinning at home (with less salty and oily food)	55.0% (360)
Increased quantity of fruit consumption	38.6% (253)
Increased frequency of consuming take-away food (more oily)	29.2% (191)
Increased consumption of soft drinks	20.2% (132)
Increased consumption of desert	19.8% (130)
Increased consumption of crispy food	19.7% (129)
Decreased consumption of water	16.9% (111)
**Self-reporting change in eating habits (Total number = 1087)**	
Reporting significant change in eating habits	17.5% (190)
Reporting modest change in eating habits	42.8% (465)

**Table 8 ijerph-18-10483-t008:** Intention to maintain precautionary measures over next three months post-test.

Precautionary measures (Number of valid cases with those missing and unsure cases eliminated)	Percentage of students (number)
Will continue to wear mask in public place (989)	92.1% (911)
Will continue handwashing meticulously (1001)	71.0% (711)
Should maintain 1-meter physical distancing (923)	37.5% (346)
Can relax 1-meter physical distancing to allow better social interaction (923)	55.5% (512)
If there is adequate space, it is not necessary to restrict number of people in gathering (903)	15.1% (136)
Can relax restriction of number of people in gathering to allow better social interaction (903)	49.3% (445)
If experiencing respiratory symptoms, will stop going to schools or activities (923)	85.8% (792)
If experiencing respiratory symptoms with no fever, will report and continue to go to school (923)	20.7% (191)
If experiencing respiratory symptoms with no fever, will report and continue to attend activities (923)	14.2% (131)

## Data Availability

Not applicable.

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
