# Peer review of "Impact of COVID-19 on Life of Students: Case Study in Hong Kong"

_ijerph, 2021, doi:10.3390/ijerph181910483_

Round 1
Reviewer 1 Report
This is a very interesting paper in which authors try to present the impact of COVID-19 on life of students in Hong Kong. However, it needs some major revisions:
1) Abstract and Methodolody: The dates of the follow-up are not consistent. (Abstract: June and July 2020 vs Methodology: June and July 2021)
2) Introduction: Last paragraph of the Introduction section should change place. This paragrpah belongs to the Methodology section.
3) Introduction: The last paragraph should present clearly the aim of the study.
4) Table 1. Location of participating schools: Are they private or public schools? Are these locations near to each other? Are the any differences between the schools or the places that should be mentioned? Economical, environmental differences?
5) The fact that the study has enrolled more girls than boys should be decribed in the limitation section.
6) Page 4 "Self-reported body weight of students was classified into .... is considered normal [26]": This paragraph belongs to the methodology section.
7) Tables, "NS": Is there a reason why authors did not provide the exact p-values?
8) Discussion, 1st-2nd paragraphs: Usually, limitation section is placed just before the conclusions. I believe that is negative for a paper to begin with the limitation of the study.
Author Response
Dr. Lorena Charrier, Prof. Dr. Paola Dalmasso ,Dr. Paola Nardone, Dr. Paola Berchialla
Guest Editors for the Special Issue Promoting Adolescent Health and Wellbeing for a Better Transition to Healthy Lifestyle Adulthood, International Journal of Environmental Research and Public Health
Ref: Impact of COVID-19 on life of students: Case study in Hong Kong Manuscript ID: ijerph-1382196
Dear Drs Charrier, Dalmasso, Nardone, Berchisalla,
Thank you very much for sending me the invaluable comments by expert reviewers. I have addressed their comments accordingly.
Reviewer 1
This is a very interesting paper in which authors try to present the impact of COVID-19 on life of students in Hong Kong. However, it needs some major revisions:
Response: Thank you for the kind words and support. We have revised accordingly.
1) Abstract and Methodology: The dates of the follow-up are not consistent. (Abstract: June and July 2020 vs Methodology: June and July 2021)
Response: We apologise for the typo and revised accordingly. It should be June and July, 2020.
2) Introduction: Last paragraph of the Introduction section should change place. This paragraph belongs to the Methodology section.
Response: The last paragraph is shifted to Methodology as 2.1 Study Design.
3) Introduction: The last paragraph should present clearly the aim of the study.
Response: Thank you for the suggestion. The last paragraph states the aim of study. “The aim of this study is to study the impact of COVID-19 on student health and well-being by collecting data reflecting the health and well-being of students at the entry of ASAP before COVID-19 outbreak) then a yearly interval (after the outbreak) to analyse any changes.”
4) Table 1. Location of participating schools: Are they private or public schools? Are these locations near to each other? Are the any differences between the schools or the places that should be mentioned? Economical, environmental differences?
Response: All the schools except one in Shatin are publicly funded. The school in Shatin is partially funded by government and partially fee paying. Sham Shui-po, Kwun Tong, Yau Tsim Mong and Kowloon City are in urban setting. Kwai Tsing and Shatin are satellite towns (urban settings). Tuen Mun and Sai Kung are semi-urban settings. Half of the students are in middle affluence group, one quarter in high affluence group and around another quarter in low affluence group. A brief explanation with footnote is included to describe the status of urban development.
Most of schools in the study are located in districts with monthly median domestic household incomes below the overall median level in Hong Kong (details are added to Table 1). The sample is not skewed towards higher socio-economic groups. This is also addressed in limitation section.
5) The fact that the study has enrolled more girls than boys should be described in the limitation section.
Response: There are 2 girl schools and others are co-education. It is addressed in the limitation section
6) Page 4 "Self-reported body weight of students was classified into .... is considered normal [26]": This paragraph belongs to the methodology section.
Response: Thank you for suggestion. This paragraph is now under the methodology section (sub-section- measuring tools).
7) Tables, "NS": Is there a reason why authors did not provide the exact p-values?
Response: The exact p-values are reported for those items not statistically significant.
8) Discussion, 1st-2nd paragraphs: Usually, limitation section is placed just before the conclusions. I believe that is negative for a paper to begin with the limitation of the study.
Response: Thank you for the suggestion. The limitation is shifted towards the end.
Please kindly refer to second file for response to reviewer 2.
I have revised the changes with marked changes. I will be most grateful for your expert editorial comments. Thank you for your kindest attention and consideration!
Best regards,
Albert Lee (Clinical Professor of Public Health and Primary Care)
Reviewer 2 Report
Dear authors,
Here are some recommendations and suggestions about your manuscript:
*Title*
Lines 2-3
Is it a “case study” or a “multicenter longitudinal cohort study”?
I think you should include a "Design section” in the methodology and explain this.
“The case study or case analysis is a research instrument or method originating from medical, psychological and educational research, which focuses on a specific case of a single person or group, which is exclusive and particular to that person or group. This design has been used in sociology by authors such as Herbert Spencer, Max Weber, Robert Merton, and Immanuel Wallerstein. It is still used today in the social sciences as a qualitative assessment method. Educational psychologist Robert E. Stake is a pioneer in its application to educational evaluation.”
*Abstract*
Lines 11-27
It is correct, it presents the synthetic information. Congratulations.
*Background and introduction*
Lines 32-36
“Since the declaration of the COVID-19 pandemic, lockdown measures have been implemented in many parts of the world. Implementation of physical measures to interrupt or reduce the spreading of respiratory viruses based on sustained physical distancing, restriction of social gathering, and “shut-down” measures has a strong potential to reduce the magnitude of the epidemic peak of COVID-19.”
Could you include at least one bibliographic reference for paragraph 1?
Lines 49-51:
“Increasing physical activity and maintaining a healthy diet leading to positive changes to their physical health has also been shown to associate with better mental health.”
Can you include the bibliographic reference?
Lines 51-53:
“Non-communicable diseases such as cardiovascular diseases, chronic lung diseases, cancer and diabetes are still constituting the main health burdens of society.”
Can you include the bibliographic reference?
Lines 57-60:
“A substantial burden of global cardiovascular disease morbidity and mortality is attributable to a sedentary lifestyle, and the attributable burden of high BMI has increased in the past 23 years.”
“Physical inactivity and unhealthy eating are 59 the key underlying causes.”
Can you include the bibliographic references for these two sentences?
Line 60:
“COVID-19 also has an impact on the day-to-day life of students with school closure.”
Can you include the bibliographic reference?
Lines 63-66
“It may need to leverage decision‐making frameworks, such as the Multi‐Tiered Systems of Support/Response‐to‐Intervention (MTSS/RTI) framework to identify needs and target instruction where it matters most when school begins in late 2020.”
Can you include the bibliographic reference or bibliographic references for this statement?
*Materials and Methods*
Suggestion: would the methods section be clearer if it included a “data collection” section and a “data analysis” section?
Lines 85-90
If "the impact of covid-19" is studied, it would be interesting to detail all the restrictive measures that the Hong Kong government took, how long they lasted, etc.
Lines 85-90 describe some details of the ASAP program.
This is confusing, since the impact of this program is not studied.
It would be interesting to make it clear that “the impact of COVID-19” on the lives of students (who received the ASAP program) is being studied.
Line 94:
Suggestion: It is indicated that the surveys were anonymous.
Could you explain the procedure for obtaining informed consent from students? Can you indicate if the study was approved by an ethics committee?
*Findings*
Line 152:
Table 1. Consider removing from the table the rows that use means and standard deviations:
“Average year ± SD at baseline (Sep to Oct 2019): 13.28 years ± 1.07”
“Average year± SD at follow up (Jun to Jul 2020): 13.99 years ± 1.07”
It is not appropriate to mix different statistics in the same table. They are presenting frequencies and percentages.
Suggestion: You can present this age data descriptively in the text.
Lines 153-160:
The weight classification corresponds to the method section, “measures section”.
“Self-reported body weight of students was classified into wasting, desirable, and 153 obese according to the weight-for-height charts in a local guide to childhood growth and 154 nutrition assessment by Leung [25].”
“The charts are gender-specific, in which obesity is defined as body weight values above 120% of the median weight-for-height while wasting is defined as body weight values below 80% of the median weight-for-height. When the body height value of a subject exceeds the data available in the charts, Body Mass Index (BMI) cutoffs for Asian adult populations are used to interpret the subject’s body weight that a value between 18.5 kg/m2 and 22.9 kg/m2 is considered normal [26]”
Line 176:
"It may be due to adaptation."
This should go into the discussion. It is a conjecture; it is not a result.
Line 184:
Table 2: I consider that the column "Number of valid cases" can be eliminated.
Suggestion:
Time spent on physical activities (number of valid cases=1081)
Types of activities engaged over last 7 days (number of valid cases=1074)
Line 202
Table 6. In Europe, influenza vaccination is only administered to at-risk populations.
In this study, the participants are young people between 11 and 15 years old.
Are these high school students the target population for the influenza vaccine in Hong Kong?
Is vaccination recommended in this population?
*Discussion and Conclusion*
It is adequate, comprehensive and well argued.
Maybe I would not present the limitations at the beginning.
Congratulations.
Author Response
Dr. Lorena Charrier, Prof. Dr. Paola Dalmasso ,Dr. Paola Nardone, Dr. Paola Berchialla
Guest Editors for the Special Issue Promoting Adolescent Health and Wellbeing for a Better Transition to Healthy Lifestyle Adulthood, International Journal of Environmental Research and Public Health
Ref: Impact of COVID-19 on life of students: Case study in Hong Kong Manuscript ID: ijerph-1382196
Dear Drs Charrier, Dalmasso, Nardone, Berchisalla,
Thank you very much for sending me the invaluable comments by expert reviewers. I have addressed their comments accordingly.
Reviewer 2
Dear authors,
Here are some recommendations and suggestions about your manuscript:
*Title*
Lines 2-3
Is it a “case study” or a “multicenter longitudinal cohort study”?
I think you should include a "Design section” in the methodology and explain this.
“The case study or case analysis is a research instrument or method originating from medical, psychological and educational research, which focuses on a specific case of a single person or group, which is exclusive and particular to that person or group. This design has been used in sociology by authors such as Herbert Spencer, Max Weber, Robert Merton, and Immanuel Wallerstein. It is still used today in the social sciences as a qualitative assessment method. Educational psychologist Robert E. Stake is a pioneer in its application to educational evaluation.”
Response: Study design section is added under methodology (section 2.1)
For this case study, comparative analyses between baseline and follow-up period were conducted to identify potential changes in students' weight status, sleep pattern and quality, the pattern of sedentary lifestyles, the pattern of physical activity, attitudes and perceived barriers for exercise, and hand hygiene. The attitudes toward precautionary measures to COVID-19 and influenza vaccination, self-reported changes of attitudes and beliefs.
*Abstract*
Lines 11-27
It is correct, it presents the synthetic information. Congratulations.
Response: Thank you.
*Background and introduction*
Lines 32-36
“Since the declaration of the COVID-19 pandemic, lockdown measures have been implemented in many parts of the world. Implementation of physical measures to interrupt or reduce the spreading of respiratory viruses based on sustained physical distancing, restriction of social gathering, and “shut-down” measures has a strong potential to reduce the magnitude of the epidemic peak of COVID-19.”
Could you include at least one bibliographic reference for paragraph 1?
Response: Reference is added.
Prem K, Liu Y, Russell TW, et al. The effect of control strategies to reduce social mixing on outcomes of the COVID-19 epidemic in Wuhan, China: a modelling study. Lancet Public Health 2020; 5(5): E261-270. https://doi.org/10.1016/ S2468-2667(20)30073-6.
Lines 49-51:
“Increasing physical activity and maintaining a healthy diet leading to positive changes to their physical health has also been shown to associate with better mental health.”
Can you include the bibliographic reference?
Response: Two references have been added
- O'Neil A, Quirk SE, Housden S, Brennan SL, Williams LJ, Pasco JA, Berk M, Jacka FN. Relationship between diet and mental health in children and adolescents: a systematic review. Am J Public Health. 2014 104(10): e31-42. doi: 10.2105/AJPH.2014.302110. PMID: 25208008; PMCID: PMC4167107.
- Sharma A, Madaan V, Petty FD. Exercise for mental health. Prim Care Companion J Clin Psychiatry. 2006;8(2):106. doi:10.4088/pcc.v08n0208a
Lines 51-53:
“Non-communicable diseases such as cardiovascular diseases, chronic lung diseases, cancer and diabetes are still constituting the main health burdens of society.”
Can you include the bibliographic reference?
Response: Reference is added
- Benziger CP, Roth GA, Moran AE. The Global Burden of Disease Study and the Preventable Burden of NCD. Glob Heart. 2016 Dec;11(4):393-397. doi: 10.1016/j.gheart.2016.10.024. PMID: 27938824.
Lines 57-60:
“A substantial burden of global cardiovascular disease morbidity and mortality is attributable to a sedentary lifestyle, and the attributable burden of high BMI has increased in the past 23 years.”
“Physical inactivity and unhealthy eating are 59 the key underlying causes.”
Can you include the bibliographic references for these two sentences?
Response: GBD 2013 is cited.
- GBD 2013 Risk Factors Collaborators. Global, regional, and national comparative risk assessment of 79 behavioural, environmental and occupational, and metabolic risks or clusters of risks in 188 countries, 1990–2013: A systematic analysis for the Global Burden of Disease Study 2013. Lancet 2015, 386, 2287-2323, doi: 10.1016/S0140-6736(15)00128-2.
Line 60:
“COVID-19 also has an impact on the day-to-day life of students with school closure.”
Can you include the bibliographic reference?
Response: Reference is added
- Bin Nafisah S, Alamery AH, Al Nafesa A, Aleid B, Brazanji NA. School closure during novel influenza: a systematic review. J Infect Public Health 2018; 11: 657–61.
Lines 63-66
“It may need to leverage decision‐making frameworks, such as the Multi‐Tiered Systems of Support/Response‐to‐Intervention (MTSS/RTI) framework to identify needs and target instruction where it matters most when school begins in late 2020.”
Can you include the bibliographic reference or bibliographic references for this statement?
Response: Reference is added
14.McIntosh, K., & Goodman, S. (2016). Integrated multi- tiered systems of support: Blending RTI and PBIS. New York, NY: Guilford, 2016
*Materials and Methods*
Suggestion: would the methods section be clearer if it included a “data collection” section and a “data analysis” section?
Response: Sections of data collection and data analysis are added.
Lines 85-90
If "the impact of covid-19" is studied, it would be interesting to detail all the restrictive measures that the Hong Kong government took, how long they lasted, etc.
Response: Thank for reminding us. The impact of COVID-19 on schools is added under 5th paragraph under background and introduction following the description of impact of COVID-19 on schools.
During the first half of academic year in 2020 in Hong Kong, the schools were closed during the spring term with online learning with half day session in summer term before closure again due to third wave in July 2022. The schools re-open after summer break in September 2020 with half-day session and closed again in early December 2020 due to fourth wave. The schools re-opened in February 2021 with half-day session. During the first half of academic year in 2020, the schools were closed during the spring term with online learning with half day session in summer term before closure again due to third wave in July 2020. The schools re-open after summer break in September 2020 with half-day session and closed again in early December 2020 due to fourth wave. The schools re-opened in February 2021 with half-day session. The Government has imposed restriction of social gathering including numbers of people grouped together and operation mode of restaurants and recreation facilities. Many recreation facilities including public utilities were closed or operated under strict control of people flow from time to time in 2020. There is a need to study the impact of COVID-19 on student life with disruption of usual school life and social interaction during that period.
Lines 85-90 describe some details of the ASAP program.
Response: Thank you for advising us. It is included in second half of second last paragraph under introduction.
“The ASAP Project provided health educational materials covering nine teaching units designed for junior secondary schools. Topics covered sleep hygiene, infectious disease control, travellers’ health, physical activity, body image, stress management, etc. From which, teachers chose one or more units for school-based curriculum enrichment. They were also required to develop experiential learning activities for students based on the topics they have taught. Students might conduct project learning on them as well.”
This is confusing, since the impact of this program is not studied.
It would be interesting to make it clear that “the impact of COVID-19” on the lives of students (who received the ASAP program) is being studied.
Response: Thank you for advising us. It is added in the last paragraph of background and introduction describing the aim of the study.
Line 94:
Suggestion: It is indicated that the surveys were anonymous.
Could you explain the procedure for obtaining informed consent from students? Can you indicate if the study was approved by an ethics committee?
Response: The study has been approved by Survey and Behaviour Research Ethics Committee (SBRE-19-104). The issues of anonymous survey and consent are stated. They are described under last paragraph of study design.
*Findings*
Line 152:
Table 1. Consider removing from the table the rows that use means and standard deviations:
“Average year ± SD at baseline (Sep to Oct 2019): 13.28 years ± 1.07”
“Average year± SD at follow up (Jun to Jul 2020): 13.99 years ± 1.07”
It is not appropriate to mix different statistics in the same table. They are presenting frequencies and percentages.
Suggestion: You can present this age data descriptively in the text.
Response: They are moved and described in the text.
Lines 153-160:
The weight classification corresponds to the method section, “measures section”.
“Self-reported body weight of students was classified into wasting, desirable, and 153 obese according to the weight-for-height charts in a local guide to childhood growth and 154 nutrition assessment by Leung [25].”
“The charts are gender-specific, in which obesity is defined as body weight values above 120% of the median weight-for-height while wasting is defined as body weight values below 80% of the median weight-for-height. When the body height value of a subject exceeds the data available in the charts, Body Mass Index (BMI) cutoffs for Asian adult populations are used to interpret the subject’s body weight that a value between 18.5 kg/m2 and 22.9 kg/m2 is considered normal [26]”
Response: Thank you for suggestion. This has been moved to methodology section. Please also refer to response to reviewer 1.
Line 176:
"It may be due to adaptation."
This should go into the discussion. It is a conjecture; it is not a result.
Response: Thank you for suggestion. This part is moved to discussion.
Line 184:
Table 2: I consider that the column "Number of valid cases" can be eliminated.
Suggestion:
Time spent on physical activities (number of valid cases=1081)
Types of activities engaged over last 7 days (number of valid cases=1074)
Response: Thank you for the suggestion. We feel that it would enable readers to interpret the findings better by reporting the number of valid cases included for analysis.
Line 202
Table 6. In Europe, influenza vaccination is only administered to at-risk populations.
In this study, the participants are young people between 11 and 15 years old.
Are these high school students the target population for the influenza vaccine in Hong Kong?
Is vaccination recommended in this population?
Response: influenza vaccination is also recommended among students in Hong Kong. This is added in paragraph before table 6.
*Discussion and Conclusion*
It is adequate, comprehensive and well argued.
Maybe I would not present the limitations at the beginning.
Congratulations.
Response: Thank you for the encouragement. The limitations are moved toward end of discussion.
I have revised the changes with marked changes. I will be most grateful for your expert editorial comments. Thank you for your kindest attention and consideration!
Best regards,
Albert Lee (Clinical Professor of Public Health and Primary Care)